# Using topic modeling to detect cellular crosstalk in scRNA-seq

**Alexandrina Pancheva**[1]*, **Helen Wheadon**[2], **Simon Rogers**[3], **Thomas D. Otto**[1]*

**1** Institute for Infection, Immunity and Inflammation, University of Glasgow, Glasgow, United Kingdom, **2** Institute of Cancer Sciences, University of Glasgow, Glasgow, United Kingdom, **3** School of Computing Science, University of Glasgow, Glasgow, United Kingdom

* a.pancheva.1@research.gla.ac.uk (AP); ThomasDan.Otto@glasgow.ac.uk (TDO)

## Abstract

Cell-cell interactions are vital for numerous biological processes including development, differentiation, and response to inflammation. Currently, most methods for studying interactions on scRNA-seq level are based on curated databases of ligands and receptors. While those methods are useful, they are limited to our current biological knowledge. Recent advances in single cell protocols have allowed for physically interacting cells to be captured, and as such we have the potential to study interactions in a complemantary way without relying on prior knowledge. We introduce a new method based on Latent Dirichlet Allocation (LDA) for detecting genes that change as a result of interaction. We apply our method to synthetic datasets to demonstrate its ability to detect genes that change in an interacting population compared to a reference population. Next, we apply our approach to two datasets of physically interacting cells to identify the genes that change as a result of interaction, examples include adhesion and co-stimulatory molecules which confirm physical interaction between cells. For each dataset we produce a ranking of genes that are changing in subpopulations of the interacting cells. In addition to the genes discussed in the original publications, we highlight further candidates for interaction in the top 100 and 300 ranked genes. Lastly, we apply our method to a dataset generated by a standard droplet-based protocol not designed to capture interacting cells, and discuss its suitability for analysing interactions. We present a method that streamlines detection of interactions and does not require prior clustering and generation of synthetic reference profiles to detect changes in expression.

## Author summary

While scRNA-seq research is a dynamic area, progress is lacking when it comes to developing methods that allow analysis of interaction independent of curated resources of known interacting pairs. Recent advances of sequencing protocols have allowed for interacting cells to be captured. We propose a novel method based on LDA that captures changes in gene expression as a result of interaction. Our method does not require prior information in the form of clustering or generation of synthetic reference profiles. We

**Data Availability Statement:** All relevant data are within the manuscript and its Supporting information files. Code available on github: https://github.com/alexpancheva/ldapaper.

**Funding:** AP is supported by MRC grant number: MR/N013166/1. TO is supported by the Wellcome Trust grant 104111/Z/14/ZR The funders had no role in study design, data collection and analysis, decision to publish, or preparation of the manuscript.

**Competing interests:** The authors have declared that no competing interests exist.

demonstrate the suitability of our approach by applying it to synthetic and real datasets and use it to capture biologically interesting interaction candidates.

This is a *PLOS Computational Biology* Methods paper.

## Introduction

In recent years single cell RNA-seq (scRNA-seq) has gained immense popularity as it provides the capability to study rare cell types, heterogeneity of cell populations, and disease and developmental trajectories [1].

Cell-cell communication is vital for most biological processes, from maintaining homeostasis to determining specific immune responses [2]. In disease states, malfunctioning cells can induce changes in cell-cell interactions and secondary changes in their micro-environment, which leads to reprogramming of the niche to their advantage [3]. Improving understanding of essential interactions has the potential to aid discovery of novel therapeutic targets [4].

One way to study interactions between cell types, widely used in scRNA-seq studies, relies on ligand-receptor pairs screenings. Examples of such methods, using a priori curated interactions include: CellPhoneDB, NicheNet, and SingleCellSignalR [5–7]. CellPhoneDB or variations of their method have been applied in practice to answer questions about intercellular communication between cell types in a range of tissues. For example, Cohen et al [8] consider the interaction of lung basophils with the immune and non-immune compartment by examining known ligand-receptor pairs and how those potentially link to development. As these methods are based on databases of curated resources, they do not allow for new genes that change as a result of interaction between cell types to be identified, so results are limited to known biology. Furthermore, most curated resources of ligand-receptor pairs are only available for humans or mouse orthologs [5].

In scRNA-seq, it is possible for two cells to be sequenced together, known as "doublets". Often doublets are a result of errors in cell sorting or capture, but recently two studies have shown that doublets can capture two physically interacting cells (PICs), offering a valuable method to measure the transcription pattern of interaction, without relying on prior knowledge. Boisset et al [9] used mouse bone marrow (BM) to demonstrate that cell-cell interaction can be studied by dissociating physically interacting doublets. The two interacting cells are separated by needles and sequenced. Further experiments also managed to infer interactions by sequencing intact doublets which were then deconvoluted based on the gene expression of the sequenced singlets. Giladi et al [10] developed a method for sequencing PICs, known as PIC-seq. With other single cell technologies, information about cell-cell interactions are lost due to cell dissociation while PIC-seq captures pairs of interacting cells. PICs are isolated by a combination of tissue dissociation, staining for mutually exclusive markers, and flow cytometry sorting. Single positive and PIC populations are then sequenced. The ability to capture PICs allows Giladi et al [10] to study physical interactions between cells and potentially capture a novel set of genes that might be changing as a result of physical proximity. On the computational side of their PIC-seq approach, they [10] cluster the mono-cultures and from these the gene expression of each PIC is modeled as a doublet: $\alpha * A + (1 - \alpha) * B$, where $A$ and $B$ are the two cell types that make the PIC and $\alpha$ is the mixing parameter. $\alpha$ is estimated by a linear regression

model trained on synthetic PICs. This is followed by maximum likelihood estimation (MLE) of $A$ and $B$. By identifying the two subtypes that comprise the PIC, expected expression can be computed. Expected and actual expression of the PIC are compared to identify changes as a result of interaction [10]. There are several potential limitations of the outlined approach. Since the PIC-seq algorithm relies on deconvoluting doublets, it cannot be applied to transcriptionally similar cells, such as subtypes or the same cell type. Furthermore, for the training of the linear regression, synthetic PICs are created by pairing pooled cells from $A$ and $B$ that are then downsampled to a predefined total number of unique molecular identifiers (UMIs). While the approach of combining cell profiles to create a doublet has been used with some modifications in a range of studies [11, 12], it simplifies how a doublet arises in practice [13]. Additionally, the method described in PIC-seq requires prior clustering of cells before simulating artificial PICs and deconvolution.

Initially developed for text, LDA has been applied to different types of omics data [14]. In single cell, LDA has been applied to RNA-seq, ATAC-seq, and Hi-C data. In the context of LDA, cells are equivalent to documents and genes (regions in ATAC-seq and locus-pairs in Hi-C) are words. Topics can be described as groups of genes whose expression co-varies [15–17]. The identified topics can be interpreted as general, cell type specific, or linked to technical quality of the samples. For example, ribosomal or mitochondrial-dominated topics might correspond to dying cells. In addition to the standard implementation of LDA, work has been completed to allow for simultaneous topic identification and cell clustering [18]. Furthermore, CellTree [19] uses LDA for trajectory inference: the method takes LDA in its standard form but computes chi-square distance between cells, and uses the distance to build a tree to describe a branching process. Most recently, a modified version of LDA has been used to decontaminate counts from ambient RNA: DecontX assumes counts come from two topics, native counts and ambient RNA. Using only native counts improves clustering and downstream analysis [20].

In this article we propose a novel method based on LDA that allows for identification of genes that change their expression as a result of cell-cell interaction. Once trained on a reference population we can fit LDA on an interacting population and capture changes that cannot be explained by the initially learned topics. Firstly, we show how the proposed model behaves when fit on synthetic doublets with some upregulated genes. We also show new topics are needed to model the counts of genes related to interaction, even if they are not expressed in all interacting cells. We fit LDA as described by Blei [21] on a population of singlets or sorted cells. Then we fix the topics from the singlets reference population and fit another LDA on the interacting cells population. The second LDA allows us to rank the genes that have high probabilities in the new topics. We apply our method to two datasets containing PICs and identify genes that change their expression as a result of interaction between cell types. Examples of genes include adhesion and co-stimulatory molecules, which are direct evidence of physical interaction between cells. Finally, we demonstrate the challenges of applying our method to a 10x Chromium dataset bronchoalveolar lavage fluid (BALF) of patients with COVID-19 [22]. We link our findings to how well the sequencing protocol can preserve interaction, and to what extent we can identify reference populations. However, as the work of [9] and [10] has shown, there is a need to modify currently available scRNA-seq protocols to allow physical interactions to be captured. To our knowledge, this is the first paper that models interaction using an LDA-based approach. Furthermore, our approach does not require prior clustering or synthetic generation of doublets compared to the computational approach previously used to identify genes related to interaction in the work of Giladi et al [10]. We use the genes identified by Giladi et al [10] as the ground truth and show how the number of top genes we select affects true positive and false positive rates.

In addition to identifying genes discussed by the original paper, we provide a comprehensive ranking of further genes that might change as a result of interaction, such as ones involved in cellular response and adhesion. Taking the top 5 genes per topic in the PIC-seq data allows us to identify 20 further known genes related to cell adhesion and immunity. Additionally, our ranked list of genes includes genes lacking comprehensive annotation and as such allows us to go beyond known interactions. While the analysis of Boisset et al [9] does not consider specific genes that would change as a result of interaction but focuses on cell types known to interact, we perform a literature survey to verify whether we can identify known genes related to interaction in the bone marrow. Specifically we consider top 25 genes per topic and we identify over 90 genes linked to cell adhesion and response.

## Materials and methods

### Latent Dirichlet allocation

Our approach is based on LDA, originally developed by Blei [21] that allows data to be explained by unobserved, latent groups. Those latent groups are called topics and each topic is a multinomial distribution over a set vocabulary. A document can contain words sampled from different topics. This assumption overcomes the limitations of standard mixture models where each document is generated by a single mixture component. As such, LDA allows for documents and biological data to be modeled in a less restrictive way. LDA also allows for different document-topic proportions.

In the context of scRNA-seq, cells correspond to documents and genes to words. Word frequencies are replaced by counts. We obtain a set of topic distributions over cells and a per-topic gene distribution. Given D cells (indexed d = 1,...,D), N genes (indexed n = 1,...,N), and K topics, we can define the generative process as follows:

$$
\begin{aligned}
\boldsymbol{\phi_k} &\sim Dir(\boldsymbol{\beta}), k = 1 \ldots K \\
\boldsymbol{\theta_d} &\sim Dir(\boldsymbol{\alpha}), d = 1 \ldots D \\
z_{dn} &\sim Multinomial(\boldsymbol{\theta_d}) \\
w_{dn} &\sim Multinomial(\boldsymbol{\phi_{z_{dn}}})
\end{aligned}
\tag{1}
$$

where $\boldsymbol{\alpha}$ and $\boldsymbol{\beta}$ are vectors of lengths K and V, where V is the size of the vocabulary (all genes in the dataset). $\boldsymbol{\alpha}$ and $\boldsymbol{\beta}$ are the parameters defining the Dirichlet priors over document-topic and topic-word multinomials. They control the sparsity of the model.

In our proposed approach, we initially fit LDA on a reference population: co-cultures of the cell types in the PIC-seq dataset, sorted BM cells in Boisset's dataset [9], and what we identify as singlets in the COVID-19 BALF data [22]. The assumptions of LDA fit well in the context of scRNA-seq as at any given point we can observe multiple processes in a cell. A cell can be described as a contribution from multiple topics, some specific to a cell type and some shared across all cells. Those processes can be described as genes that co-vary. As words can be in multiple topics, genes can be part of multiple processes. By fitting LDA on the co-culture of T-cells and the co-culture of dendritic cells (DCs), we obtain for each topic a distribution over genes that we then fix before we fit another LDA on the population of PICs, dissociated BM doublets, or DoubletFinder identified doublets respectively for the three datasets discussed in the results. The initial LDA captures a reference state of cells, a state without interaction. Fixing topic-gene probabilities learned from the reference, not interacting populations, allows us to capture in the new topics any changes as a result of interaction due to the setting of the datasets analysed. Our LDA approach is shown in Fig 1.

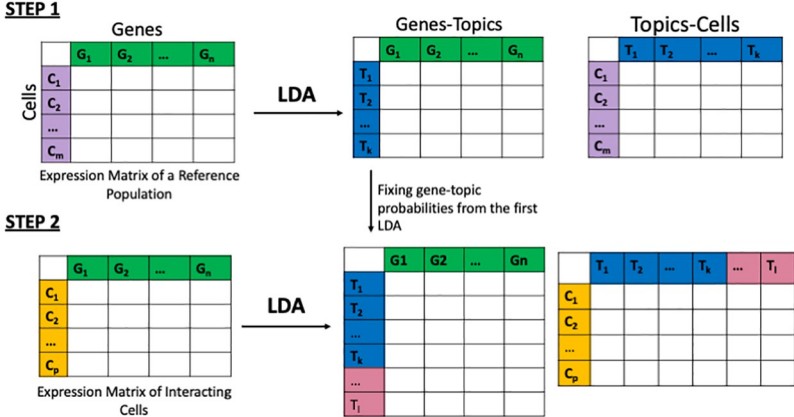

**Fig 1. We begin by fitting LDA on a reference population.** Depending on the setting of the experiment the reference could be sorted cells (Boisset's dataset) or co-cultures of the cell types involved in the interaction (PIC-seq dataset). Using the topic-gene probability vectors we learn from this first LDA, we fit another LDA with some topic-gene probabilities fixed on the interacting population.

We use variational inference to estimate the parameters of the posterior distribution of LDA. As LDA has been extensively discussed in literature, an explanation of the inference procedure can be found in [21].

## Identifying topics linked to a cell type

In order to aid interpretation of the identified topics, we link topics to cell types. For each topic we group together cells from the same cell type as annotated in the reference, and perform a Mann–Whitney U test on the topic-cell vectors. Under the null hypothesis, we assume a topic has the same probability in the two cell types. To correct for multiple testing, we use the Benjamini-Hochberg procedure with $\alpha$ set to 0.05.

## Choosing number of topics

To select a suitable range of topics, we compute perplexity for a range of topics starting with 2. A lower perplexity score is an indication of a better model. We note that perplexity decreases rapidly up to K = 10 and then flattens out. We also measure cosine and Jensen-Shannon (JS) divergence which show very similar patterns (S10 Fig). Such behaviour is common when fitting LDA-style models and it is typical to choose the smallest value of K that is able to explain the data: i.e. the value of K at the point in which the perplexity flattens out. In Fig 2, for example 10 would be a suitable number of topics for that dataset. Our goal is to recover interacting genes. To demonstrate that this strategy for choosing the number of topics is appropriate for that final goal, we show the effect the number of topics has on the Area Under the ROC curve (AUC). A plot of number of topics versus AUC can be seen in S9 Fig and shows agreement with the perplexity plot: optimal results are observed for K = 10, the value at which the perplexity flattens out. While performance is relatively consistent, the ROC curves with higher number of topics show some decay.

## Motivating the need for new topics

Consider a cell, $n$, being one of the PICs. Cell $n$ decomposes into $\theta_n$ where $\sum_k \theta_{nk} = 1$ (probability distribution for cell $n$ over all $k$ topics). Some of the topics come from the reference LDA

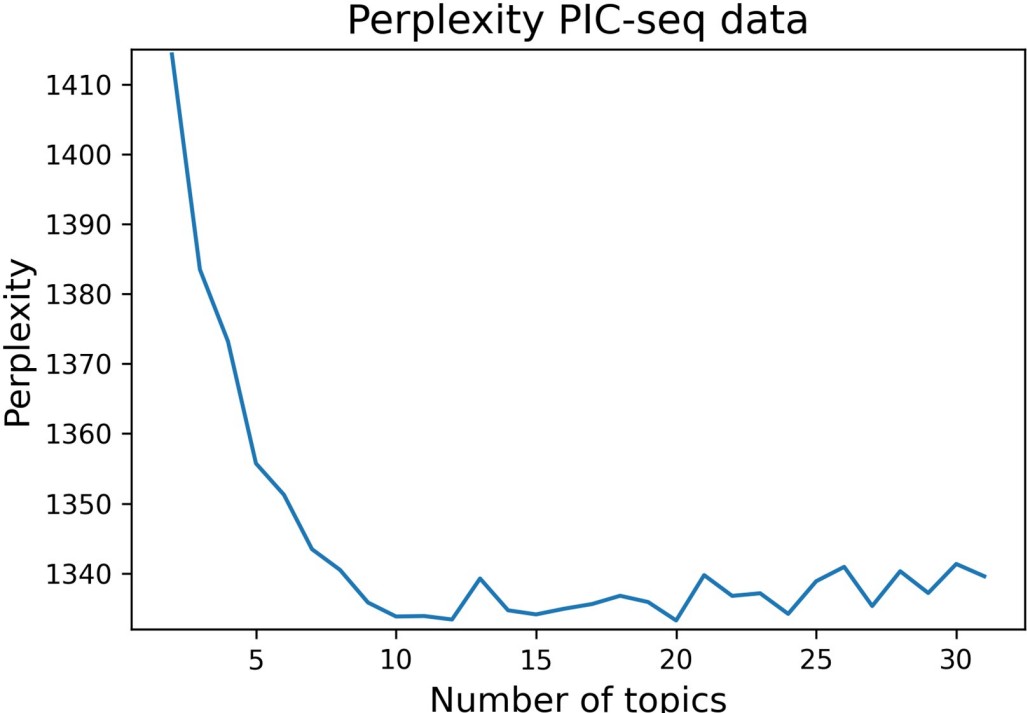

**Fig 2. Illustrating the drop of the perplexity as the number of topics increases.** In this particular case, a suitable number of topics would be 10. After that as the number of topics increases, the perplexity values plateau with some fluctuations.

fit, which are fixed before fitting the second LDA, and some topics come from the PICs. We want to compare a fit with all topics with a fit where we do not use any new topics. Let $\Delta_{nk} = \theta_{nk}$ but we set the contribution of all topics that come from the PICs to 0 and re-normalise, so that $\sum_k \Delta_{nk} = 1$. Let $\boldsymbol{\beta_k}$ = topic probability for topic $k$ and $\sum_m \beta_{km} = 1$. If we are only interested in the probability of picking the counts for one gene versus all other genes, the multinomial distribution reduces to a binomial distribution. For each topic distribution, we compute the probability $P(X >= x)$ for a binomial distribution defined as $X \sim Binomial(n, p)$ where $n$ is the total counts for cell $n$, $p$ is the probability for that gene in that topic, and $x$ is the count for a particular gene in the current cell. Once we have computed the probability for a gene for each topic, we multiply them by $\theta_{nk}$ and $\Delta_{nk}$, topic weightings for cell $n$, and sum on $k$. We expect the probability of observing counts greater than or equal to the actual count for a gene changing as a result of interaction to be higher under $\theta_{nk}$ where all topics are included compared to $\Delta_{nk}$ which is based only on the initial topics. Probabilities should be similar for genes not involved in interaction.

## Ranking genes as potential candidates of interaction

One of the outputs from our LDA is a probability distribution for a gene in a cell across all topics. Let $N$ be the total number of cells in our doublets/interacting population of interest, then for a gene $G$ and a topic $k$ we find how many cells require topic $k$ to explain the expression of gene $G$. For a newly identified topic $k$, for each gene we count how many cells have highest probability for this gene in topic $k$. For each topic, we produce a ranking of genes based on the number of cells that require this topic to explain the expression of that gene. The rankings for

each newly identified topic can then be analysed. We choose a different number of top genes from each topic in subsequent experiments. For our synthetic experiments, we plot how those values affect true and false positive rates.

## Evaluation datasets

- PIC-seq of T-cells and DCs: The count matrix and the metadata were downloaded from the Gene Expression Omnibus (GEO) under accession number GSE135382. The metadata file was used to filter for co-cultures of the same cell type and co-culture of T-cells and DCs. The reference population consists of cells tagged in the metadata as: Co-culture *TCRb*+ (T-cells) and Co-culture *CD*11*c*+ (DCs). All three timepoints 3h, 20h, and 48h were used. PICs were selected from the metadata as: Co-culture, *TCRb* + *CD*11*c*+, all three timepoints 3h, 20h, and 48h.

- BM dataset: The count data was acquired from GEO under accession number GSE89379. The sorted cells were used as a reference and the dissected doublets were used for analysing interactions. Cells prefixed JC20 to JC47 denote micro-dissected cells. Cells with prefix JC4 denote sorted hematopoietic stem cells (used as reference).

- COVID-19 BALF dataset: The data were downloaded from GEO, under accession number GSE145926, in the form of h5 files, CellRanger output. The next subsection describes how the reference and the population of potentially interacting cells are identified.

## Pre-processing and analysis before LDA

**PIC-seq and BM dataset.** PIC-seq. Since the PIC-seq dataset was generated using the MARS-seq platform which has higher sequencing depth than the standard 10x Chromium, we set higher filtering cutoffs for the number of unique features per cell. As we are not relying on clustering, we can also set a higher cutoff for the number of cells in which a gene is captured. Genes appearing in fewer than 200 cells were filtered out. Cells with more than 500 features were retained for downstream analysis. Similar to the original publication we exclude ribosomal genes.

**BM dataset**. This dataset has been sequenced using CEL-seq. Only genes expressed in more than 10 cells were considered for downstream analysis, resulting in over 10 000 total genes. 369 sorted cells and 1546 dissected doublets were used. No other pre-processing was performed before fitting LDA.

For both datasets filtering steps are performed independently of any scRNA-seq pipeline.

**COVID-19 BALF.** Quality control, filtering, normalisation, integration, and clustering were done in R, using Seurat, version 3.1.2. Filtering decisions are dataset dependent and are based on three main metrics: number of genes per cell, number of cells a gene is expressed in, and fraction of mitochondrial genes. It is typical to filter for cells with a very high number of genes expressed to prevent including doublets in the data. For example, cells with low counts and high mitochondrial fraction indicate the mRNA has leaked out through a broken membrane. As such, samples were filtered for cells with fewer than 500 genes. Since we are interested in doublets, which are often assumed to have higher counts than singlets, the maximum cut-off was relaxed [23]. To exclude dying cells we also set a mitochondrial gene expression cut-off to 25. The full list of filtering cut-offs for the different COVID-19 samples can be found in S1 Table.

Seurat's NormalizeData and FindVariableFeatures functions were used before integration. Samples were integrated first by condition, and then all conditions were integrated using FindIntegrationAnchors. Clustering resolution was set to 0.5 to identify general populations. Based on the cluster identification, a subset of the BALF cells were taken forward for LDA analysis.

To confirm cells we identified as doublets, we use DoubletFinder. Since DoubletFinder can only be used on a single sample and not integrated data, patient samples C143 and C145 were analysed separately [11]. Those two samples were chosen based on the amount of cells in what we defined as a doublet cluster. S1 Fig shows the annotated clusters for sample C145.

## Results and discussion

### Validation using synthetic doublets

Before testing our method on a real dataset of interacting cells, we apply it to a dataset containing synthetic doublets in order to show that we are able to detect genes that change in interacting cells. We simulate doublets by merging the expression profiles of singlets using different ratios: 50/50 (equal contribution of each cell type), 60/40, and 30/70. In order to obtain a ground truth for genes that change as a result of interaction, we modify the expression of some genes by adding 1.5, 3 and 10 to their total counts. Results are shown in Figs 3 and 4 and the supplementary information, S2, S3 and S6 Figs. The value of 1.5 increase was chosen as it represents a typical count for a gene. We chose to modify the following genes in the synthetic doublets: *Sell, Mif, Bcl2l1, Cd40, Myc, Ncl, Cst3, Ly6a, Ctla4, Ccl22, Cd69, Dll4, Lgals1*. We trained our first LDA on a randomly sampled subset of T-cells and DCs mono-culture from the [10] paper. After fixing the topic from the reference, we fit a second LDA on the doublets that were created as different contributions of the singlets and upregulation of some genes. We expect those genes to require contributions from the new topics to model their counts.

We applied our approach to the datasets of upregulated doublets. We expect the probability of observing counts greater than or equal to the actual counts of the list of upregulated genes (e.g. *Ly6a, Sell*) to be higher when we use all topics and we compute the probability under $\theta_{nk}$, where $k$ is topics learned from singlets and simulated interacting doublets. For the genes we chose not to upregulate, the probabilities under $\Delta_{nk}$ and $\theta_{nk}$ should be similar. This is shown in

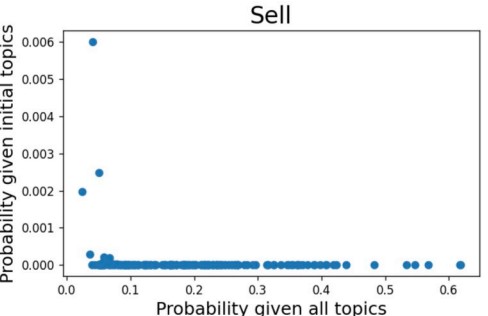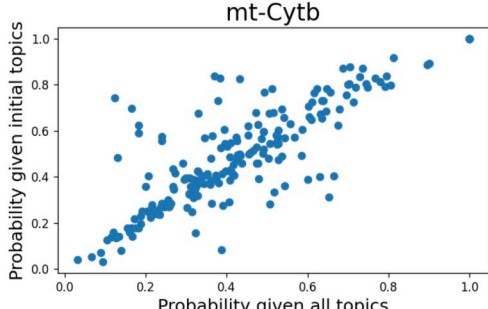

**Fig 3. Using the synthetic data, we evaluate how likely it is to observe counts for some genes under $\theta$ and $\Delta$.** For 200 upregulated doublets, we plot the probability of observing the counts of a gene that has been upregulated, *Sell*, and a gene that has not, *mt-Cytb*, under a model using all topics or using only the topics learned from the singlets populations. In the case of *Sell*, a gene with modified expression, the probability of observing the actual counts or greater than in the upregulated doublets using all topics is higher compared to the probability of observing those counts if we only use the initial topics. However, for *mt-Cytb* that we have not upregulated the probabilities under the two models of observing the actual counts or greater than are similar. Thus, we can conclude that the additional topics are required to model the genes that change.

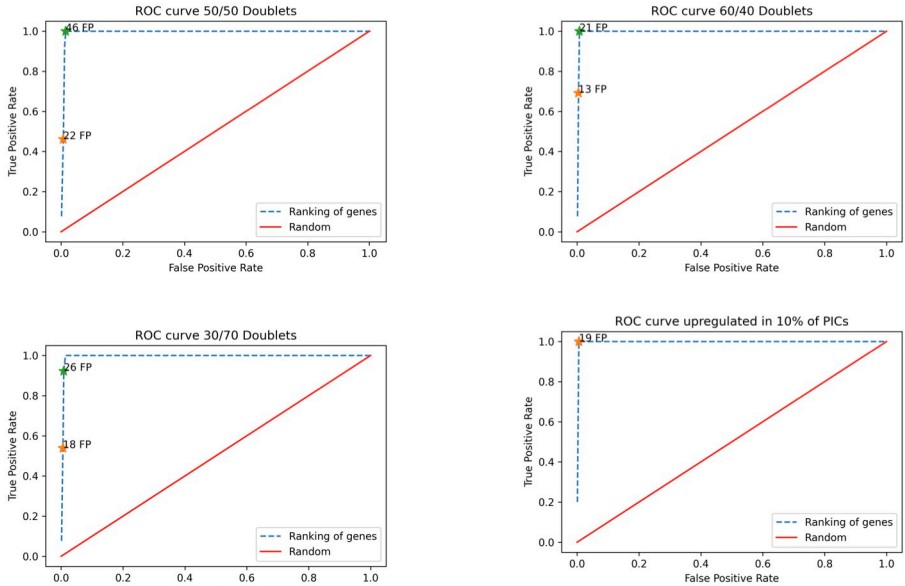

**Fig 4. For each of our 4 synthetic experiments, we plot a ROC curve using a different number of top genes based on our ranking.** In all cases the genes we have modified the expression of appear at the top of the newly identified topics. For all synthetic experiments, using up to the top 15 genes per topic resolves the list of upregulated genes we consider as ground truth. However, as seen from the plots even if a slightly higher number of top genes are used, the false positives are gradually increasing. In each plot we have indicated the number of false positive genes ranked for each experiment while the full truth set has not been identified. Total list of ranked genes for those experiments is over 2500.

Fig 3, where we plot the probability of observing counts greater than or equal to the actual counts in doublets with modified expression of the previously listed genes. *Sell*'s counts in the modified doublets can be explained better if the new topics are included. However, in the case of *mt-Cytb*, a gene we have not modified, probabilities are similar under the two models, using all topics or the initial ones. Further examples of genes we have modified and genes with counts that can be modeled by the original topics can be seen in the supplementary information S2 and S3 Figs. Results are similar to the genes shown in Fig 3.

For each of our simulated doublets experiments, we rank the genes based on whether they require contribution from the new topics to explain their expression. We plot true positive rate vs false positive rate using different cutoff values for the top ranked genes. The ROCs in Fig 4 show how the results are affected by picking a different number of top genes per topic. A further experiment was performed where we upregulated a random set of genes *Gcfc2, Wdsub1, AU040320, Pank3, Dcaf12, Gm26669, Ehd2, Bag3, Rpl10-ps2, Notch1, Ppm1g, Oxsr1, Nrarp, Ppp3ca, Rpl28-ps1, Stbd1, Srgap2, Cpped1, Gm10420, St6galnac3*. Results can be seen in the supplementary S4 and S5 Figs.

Additionally, we evaluate whether we can identify genes that change in a subset of cells by upregulating some genes in 10% of the total PICs population. We upregulate the expression of *Gbp4, Gbp7, Gzmb, Il2ra, Psma4* in 20 cells (10% of the total PICs). In all four experiments, using up to 15 top genes per topic resolves the list of upregulated genes that we use as ground truth. We note that even if a gene is upregulated in as few as 20 cells that gene can still appear in the ranking, and we recommend also exploring genes which change in few cells when analysing results.

## PIC-seq dataset

The first real dataset we use for evaluation has been generated by PIC-seq and includes interacting T-cells and DCs (gated for TCR$\beta^+$ $CD11c^+$) as well as two co-cultures of a single cell type (T-cells: TCR$\beta^+$ and DCs: CD11$c^+$) across three different timepoints, 3h, 20h, and 48h. The original [10] work uses a metacell model to cluster the cultures of a single cell type. Then each PIC is modeled as a combination of metacells, and a mixing proportion, $\alpha$, is estimated by a linear regression model trained on synthetic PICs. The metacells are identified using MLE. Genes of interest are identified by comparing expected expression, based on the inferred cell types contributing to the interacting pair and actual expression of the PICs.

Our model does not require prior clustering and generation of synthetic reference profiles. As a first step we train one LDA on the co-cultures of T-cells and DCs, using those as a reference. With the first LDA we manage to capture topics specific to T-cells and DCs (groups of genes co-varying in one cell type over the other) and specific time-points. As seen in Fig 5

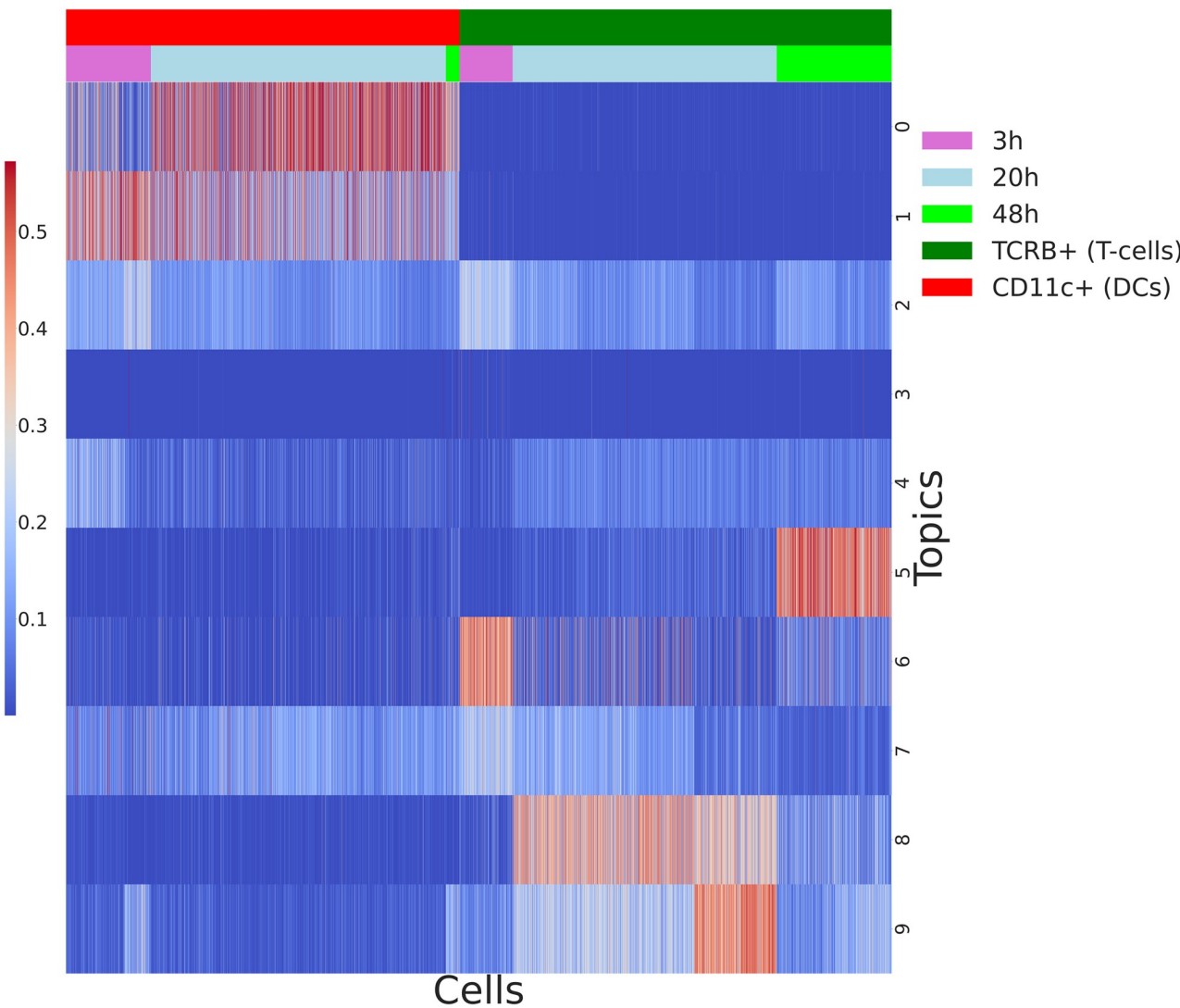

**Fig 5. Heatmap of topics expression in the reference populations of T-cells and DCs.** To map topics to a cell type, we group together cells from the same cell type and perform Mann–Whitney U test for each topic. Results are corrected using the Benjamini-Hochberg procedure for multiple testing.

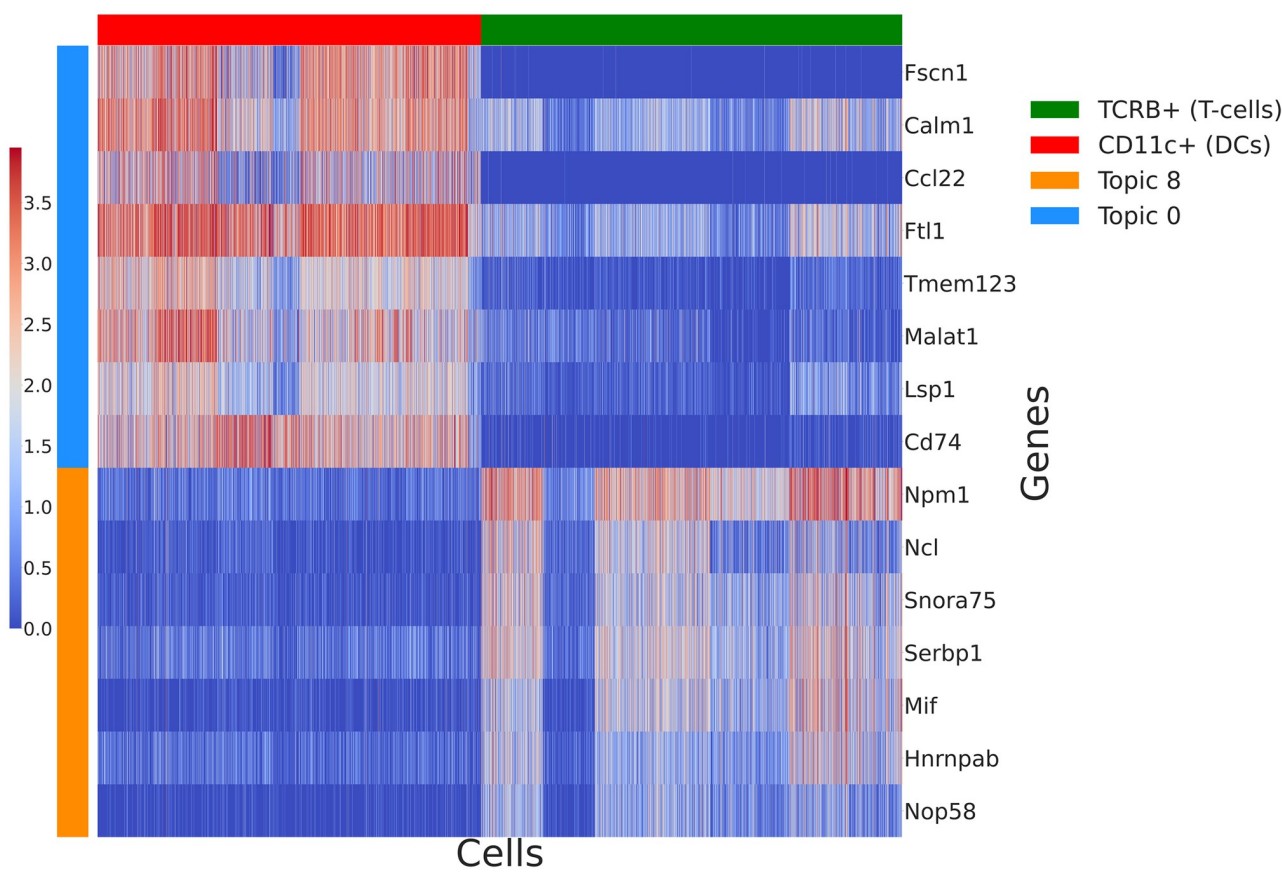

**Fig 6. Heatmap of top genes based on probability in that topic (for each topic we can obtain the probability of the genes in that topic) from topics 0 and 8 (log-transformed).** Topic 0 contains genes that tend to have higher expression in DCs compared to T-cells. Topic 8 contains genes co-varying in the T-cells co-culture. Results are similar to Figure 1 from the Giladi et al [10] manuscript.

topics 0 and 1 are specific to DCs. We identify topics specific to a cell type as described in Identifying topics linked to a cell type. In addition to having high probability for a cell type, some of those topics have higher probability over the different timepoints. For example, topic 5 has high probability for T-cells during the 48h window. Similarly topic 1 is higher for DCs in the 3h time period, while topic 0 is DC specific for the 20h time period, see Fig 5 and S3 Table. To explore what the top genes are in some of those topics, we pick topics 0 and 8 and order the genes in those topics by probability. We see DC specific genes *Fscn1, Ccl22, Tmem123,* and *Cd74* in topic 0. Similarly, some of the genes with highest probability in the T-cell specific topic include *Mif, Ncl, Nmp1* (see Fig 6). Further examples of genes with high probabilities in some of those topics can be found in S3 Table.

We fix the topics we learned from the co-cultures of the two cell types, T-cell co-culture and DCs co-culture, before we fit another LDA on the physically interacting populations of PICs. As described earlier, in order to rank interesting genes, for each topic we learned from the PICs, we count how many cells use this topic for a particular gene. Then we rank the genes based on number of cells. Our PICs population contains over 3000 cells and we filter our rankings for genes needing particular topic for fewer than 10 cells.

To validate our findings, we check whether the top genes in each of the newly learned topics have also been highlighted by Giladi et al [10] in Supplementary figure 4 of their paper. Due to

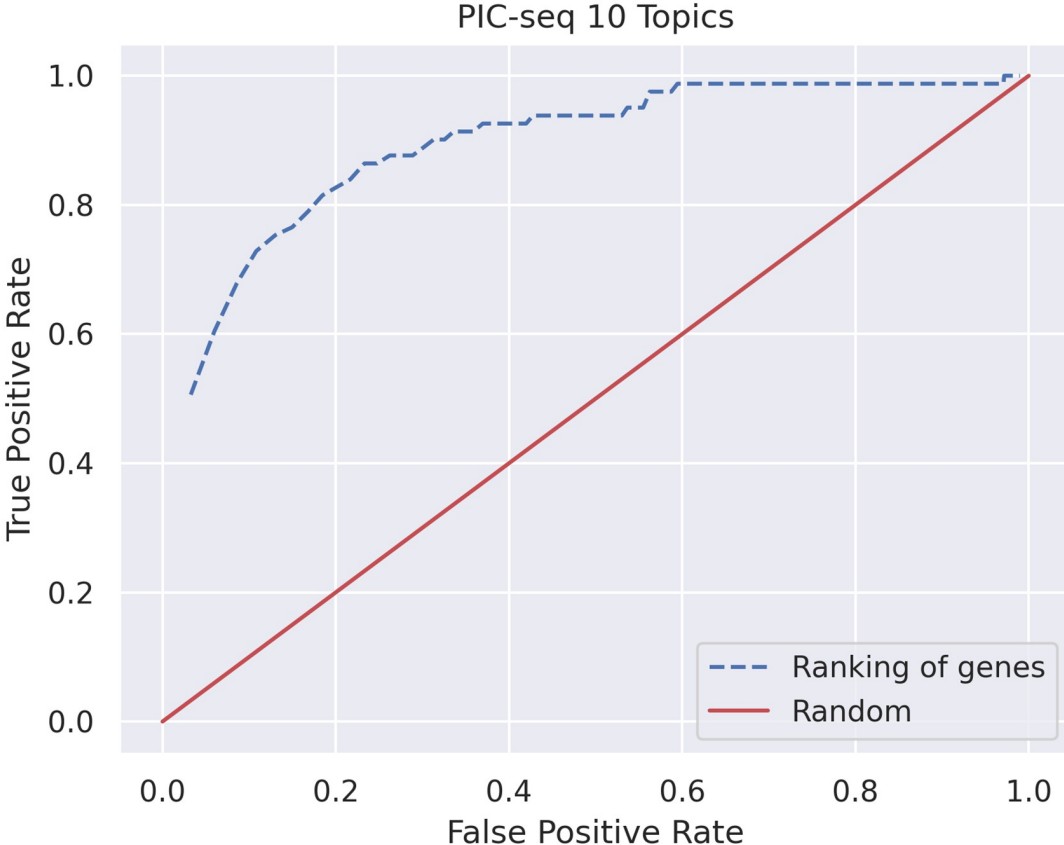

**Fig 7. ROC curve using genes from [10] as ground truth based on top genes cutoff for each topic.** It is important to note that some of our false positive values correspond to true interacting genes that have not been presented in the [10] paper amongst their Supplementary figure 4 genes.

differences in filtering, we have not retained 10 of the genes they identify to change as a result of interaction and we take the remaining 81 genes present in our data as a ground truth. In order to evaluate how results are affected by number of top genes per topic we select, we plot true positive rate vs false positive rate (see Fig 7).

While the analysis done in the original publication [10] groups cells by the types of the singlets involved in the interaction and the timepoint of capture before performing log fold change (results in Supplementary figure 4d of the original paper) we show that some of the new topics we identify correspond to the timepoints of capture and reveal genes with temporal patterns as shown in Fig 8. For example, *Ldha, Ptma, Pcna, Trac, Dut* are needed by a subset of cells and captured in the same topics. Their pattern of expression is higher after the first 3h. *Tnfrsf4, Tnfrsf9, Tnfrsf18* seem to be expressed across all timepoints and the shift of their expression is captured by the same topics. The expression of new topics across the cells can be seen in S11 Fig.

While for the purposes of the ROC analysis we are considering genes that are not amongst the ones discussed by Giladi et al [10] as false positives, for some of those genes there is evidence they could be involved in interaction. Taking the top 5 genes per topic considered as false negatives previously, we find genes related to immunity and cell adhesion, some of which are ligand-receptor pairs (over 20 genes in the first 100 ranked). Examples include *Cd2, Cd74, Il2ra, Il2rb*. Additionally, while some known genes appear high in the ranking, some of the

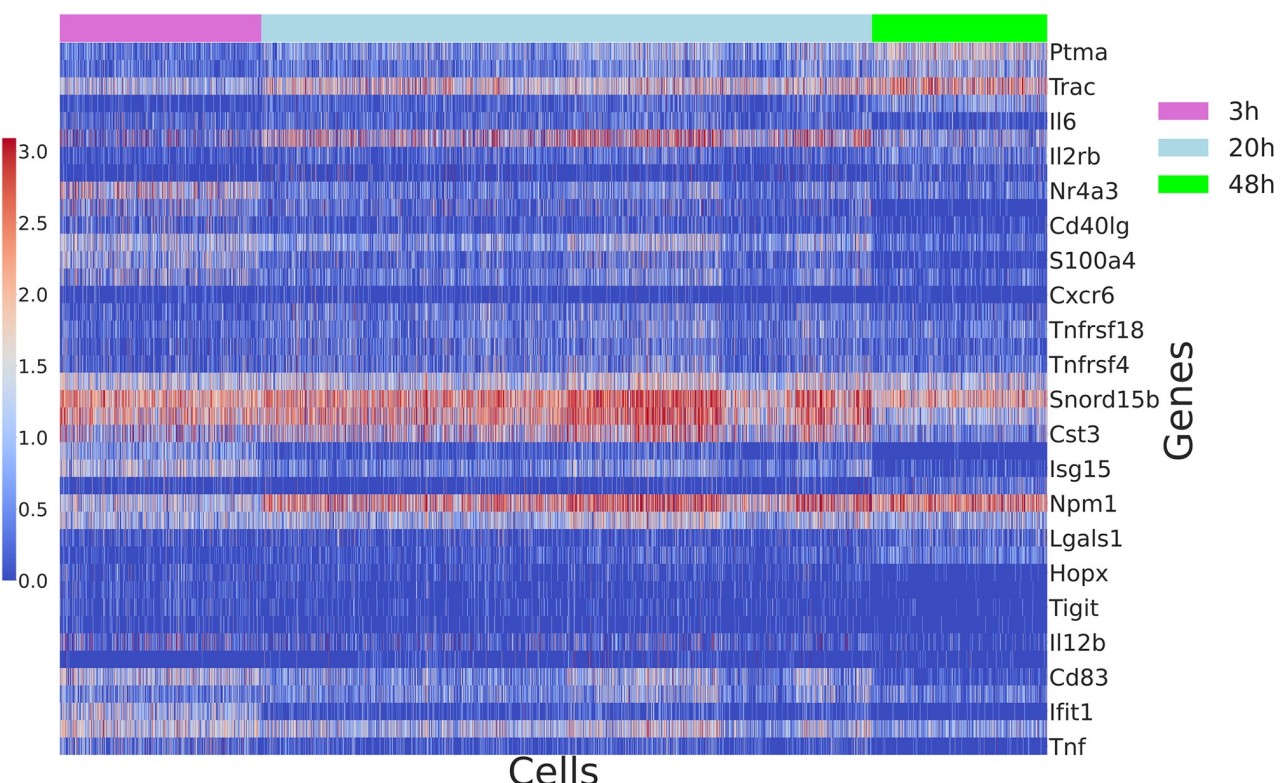

**Fig 8. Log transformed expression of genes identified by both [10] and our ranking approach (top 20) and showing the possible temporal expression pattern.** For example, genes *Ldha, Ptma, Pcna, Trac, Dut* have the highest probability in the same new topics and their expression increases after the first 3h, while *Tnfrsf4, Tnfrsf9, Tnfrsf18* do not seem to show a temporal pattern and the shift in their expression compared to the single cell type co-cultures is captured by the same topics.

genes in our list are not well-annotated. This makes them potential targets for further analysis to elucidate their role. Genes can be found in S4 Table.

## BM dataset

The original work by Boisset et al [9] is focused on identifying significant interactions between cell types, using sorted BM cells and needle dissected doublets. Firstly, we fit our LDA on the sorted BM cells and fix the learned topics. Next, we fit the second LDA on the needle dissected doublets.

We hypothesised our approach would be able to identify genes involved in the main interactions discussed by Boisset et al [9]. Their work considered three specific interacting pairs: macrophages and erythroblasts, plasma cells and myeloblasts/promyelocytes, and megakaryocytes and neutrophils. Macrophages and erythroblasts have been known to interact, and erythroblastic islands are considered an important niche for the maturation of red blood cells. In addition to anchoring erythroblasts within island niches, macrophages also provide interactions which are important for erythroid proliferation and differentiation [24]. When describing physical interactions, adhesion molecules are of particular interest. In our analysis we identify *Vcam-1* and *Itgam*, which are known to support adhesive interactions in macrophages.

Boisset et al [9] also identify and validate the interaction between megakaryocytes and neutrophils. Their findings support other studies that have looked at emperipolesis (whereby neutrophils are engulfed by BM megakaryocytes) as a process mediated by both lineages. This

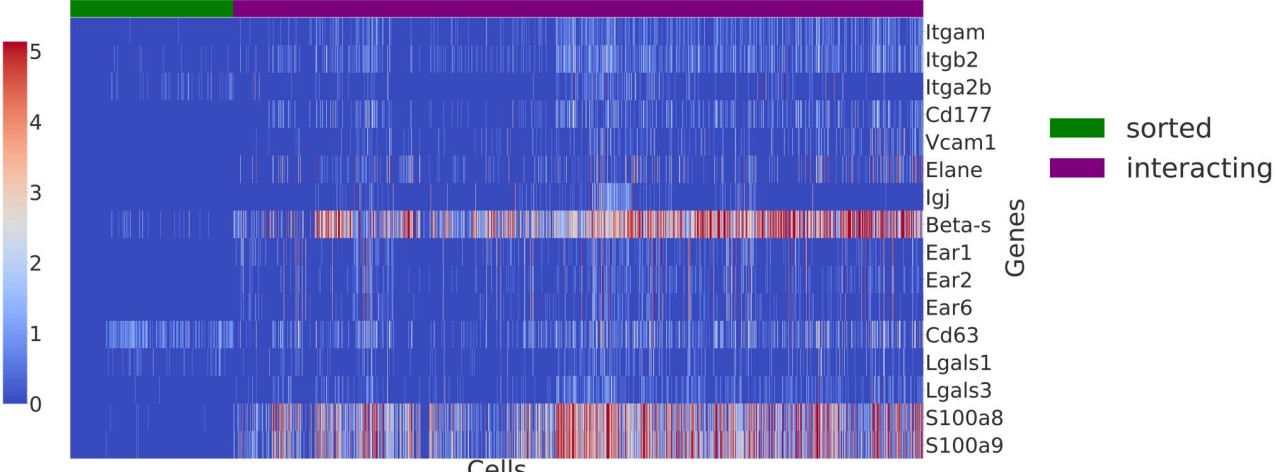

**Fig 9. Heatmap of genes chosen based on their ranking in the new topics and evidence from literature that they are linked to interactions in the BM (log-transformed).** Amongst the examples, we see genes linked to neutrophil adhesion such as *S100a8*, *S100a9*, and *Cd177*. *Elane* and *Igj* are two genes confirmed by Boisset et al [9].

interaction is important for production of platelets. Cunin et al [25] identified that emperipolesis is mediated by *β2* integrin Cd18 and Icam-1 interaction. Blocking *β2* integrin *Cd18 (Itgb2)* impairs emperipolesis [25]. This is another integrin we identify in our analysis. *Elane* and *Igj* are two other genes discussed by Boisset et al [9] that we identify to require additional topics to model their expression. The genes shown in Fig 9 are identified by taking the top 25 genes from each topic. Overall, based on top 25 genes ranking per new topic (over 300 genes in total), we identify genes linked to cell adhesion, innate immunity, and immune response. The full list of genes can be found in the supplementary information, S4 Table. While some of the genes are known to be linked to neutrophils (*Cd177, Prtn3, Serpinb1a, Lsp1*), other genes are less well-annotated in terms of function, and as such this demonstrates the benefits of using an approach that is not based on curated resources of known interactions.

## COVID-19 data

Previously, we used datasets generated by modifying standard protocols to allow for PICs to be generated. However, here we explore the potential of our method to identify genes that change as a result of interaction in datasets generated with the 10x Chromium platform, which does not have the ability to preserve interacting cells as there is no specific way of capturing doublets. We took a recently published COVID-19 BALF dataset containing several cell types like T-cells, macrophages, B-cells, DCs, and neutrophills. There are samples from patients with moderate COVID-19, severe infection, and healthy controls. During cluster annotation, the authors labeled several clusters as doublets [22]. We hypothesised that some of those doublets might represent interaction, as macrophages are known to interact with T-cells. To confirm the identity of the doublet cluster in the severe illness patient samples, we looked at marker gene expression followed by analysis with DoubletFinder.

We use the populations labeled as singlets by DoubletFinder as a reference for LDA. We fix those topics and fit a second LDA on the doublets population. Identification of potential interacting genes was performed similarly to the datasets analysed earlier, by ranking genes within each new topic based on how many cells require this particular topic to explain the gene expression. Additionally, as we only have just over 200 doublets, we only consider genes using

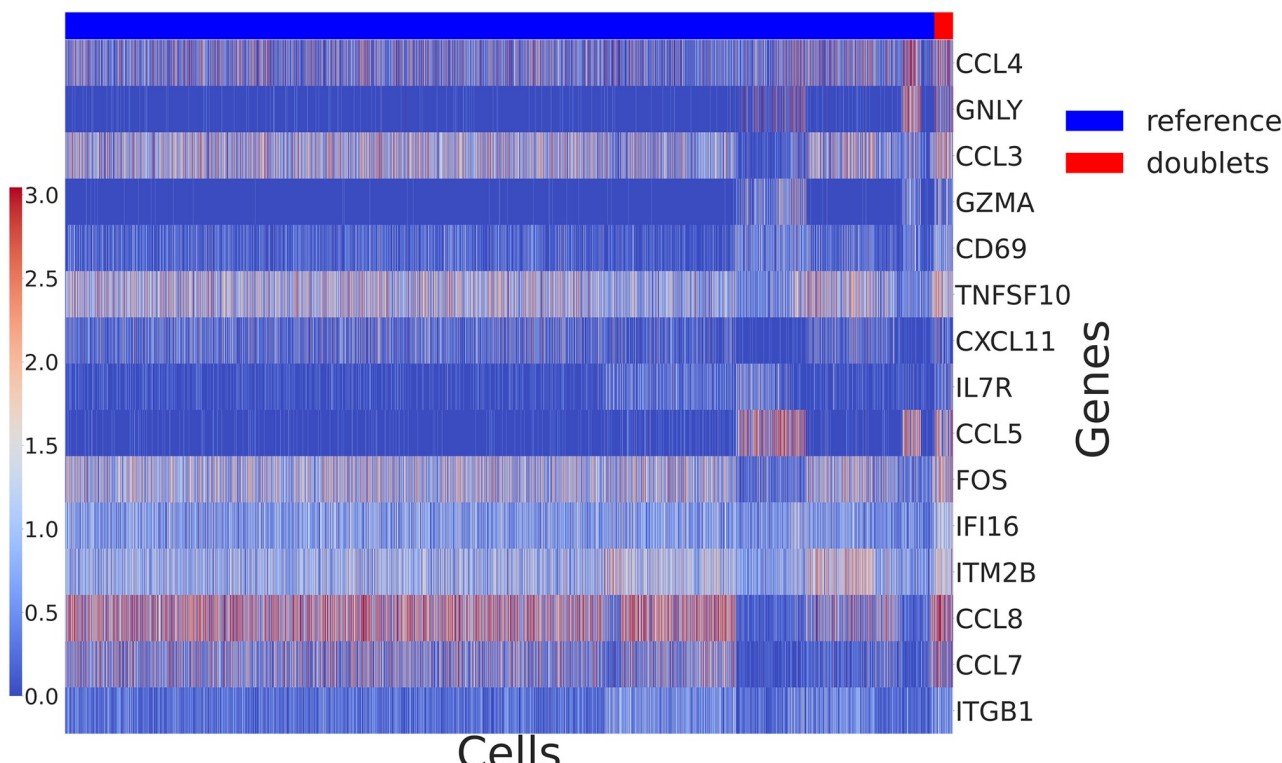

**Fig 10. Heatmap of the COVID-19 data showing the expression of top ranked genes needing new topics for a subset of the doublets population (log-transformed).** As none of the genes are uniquely expressed in the doublets population, we are capturing a shift in the expression of those genes. Some doublets can represent biologically interacting cells as the top ranking genes include cytokines and chemokines. However, due to the quality of the reference population our results are inconclusive.

a specific topic in at least 10 cells. As we can see from Fig 10, some of the genes that require contribution from the new topics to model their expression include cytokines and chemokines, which might suggest interaction. A subset of the cells also seem to require new topics for genes related to interaction. We refer to work by Takada et al [26] and Magee et al [27] that discuss physical interactions to identify suitable candidates. As not all doublets require new topics, it is possible we have a mix of interacting and technical doublets. While we are capturing a shift in the expression of certain genes, our results are inconclusive, potentially due to the quality of our reference population as the reference is constructed based on the cells DoubletFinder annotated as singlets, and as such the reference might contain some interacting cells. Depending on the amount of cells loaded, a standard 10x Chromium experiment can result in the formation of over 0.7% technical multiplets with doublets being predominant. While with DoubletFinder we have managed to label some of the doublets, computational tools for doublet detection do not achieve perfect sensitivity and specificity scores, so it is possible the reference population contains cells that exhibit signs of interaction and changes in the expression pattern of some potentially interesting genes. Our approach can identify genes that change as a result of interaction and is suitable for datasets where the reference population is clearly labeled as in the PIC-seq and dissociated BM examples discussed earlier.

## Conclusion

In this manuscript we have demonstrated the suitability of LDA for analysing PICs. We have shown that our model is sensitive to changes in gene expression that cannot be explained by

the non interacting populations and thus new topics are needed to model the expression of genes that change as a result of interaction. Our model has been applied to two datasets of sequenced PICs and a dataset generated by standard 10x Chromium. Our approach assumes there is a reference population that can be used to fit the first LDA; for example this could be populations before an interaction has occurred. In addition to genes known to be involved in interaction and discussed by [9] and [10], we also rank further candidates for interaction that might be of interest for validation. We demonstrate the challenges of applying our approach to a dataset where a reference population cannot be clearly labeled in the case of the COVID-19 BALF analysis using the standard 10x Chromium protocol. However, amongst the top genes we rank there are cytokines and chemokines, genes known to be regulated by physically interacting cells, which might suggest the doublet population includes both technical and interacting doublets. While this is informative, the current setup of the 10x Chromium protocol is not fully suitable for studying cellular crosstalk of physically interacting cells, and as such the likes of PIC-seq should be considered when studying interactions.

As seen from the datasets discussed here, modeling interactions based on doublets can be very useful. As such, distinguishing technical from biologically significant doublets poses an interesting challenge. While we have applied our LDA approach to PICs, there is potential for our work to allow for distinguishing technical doublets from transitioning cells as long as the transition is described by a unique set of genes, so that a new topic can be defined. While there are cell hashing methods that allow for mitigation of batch effects and overloading of sequencers, those methods help identify doublets between different samples/patients while intra-donor doublets remain undiscovered. On the computational side of doublet detection, methods make a range of assumptions that pose challenges to using them in practice. For example, DoubletFinder requires doublet rate which is not available in all experiments. The accuracy of all methods is affected when applied to transcriptionally similar cells, and DoubletDecon would not allow a doublet cluster to be present in the data. DoubletDecon is the only method able to distinguish technical doublets from transitioning cells. As such, there is a clear need for methods that can eliminate technical noise, but not at the expense of biological significance [11, 12, 28–30].

As PIC-seq is a very powerful approach, it can potentially be used to generate data including physically interacting doublets as well as singlets. It would be of interest to identify whether some of the singlets in fact show signs of interaction. Are they cells that have interacted but separated? Or maybe they have not interacted at all? Such datasets could easily be analysed following the methodology described earlier. We would expect to see signs of interacting topics in some singlets but maybe not all and as such we should be able to distinguish between singlets and singlets that have interacted before.

As there is a demand for understanding PICs, we believe methods like PIC-seq will be used more often in future and further work will be done to develop sequencing protocols that allow for capturing physical interactions that have the potential to become therapeutic targets. When such datasets are generated, there should be techniques that allow for their analysis and are not limited to knowledge captured in biological databases. The method described here is one such example that does not require any prior information such as clustering of the cell types involved and generation of synthetic reference profiles. As further datasets are generated, fields that would benefit from more in depth understanding of interactions include: understanding parasite-host interactions, crosstalk between immune cells and other lineages, and effect of cell-cell interaction in cancer progression.

## Supporting information

**S1 Table. Filtering parameters used for each sample in the COVID-19 dataset.** Upper cutoffs for nFeatures has been set to relatively high values as we are interested in potential doublets. The percentage of mitochondrial genes (% mt) cutoff allows us to exclude dying cells.
(PDF)

**S2 Table. We perform stage 2 by fixing these 5 topics and then fitting the second LDA on the interacting DCs and T-cells.** We observe genes related to housekeeping and mitochondrial processes. These processes also exist in the reference population but, due to the low number of topics that we specified initially, it seems that they are only picked up in the second stage. Indeed, when 10 topics are used for the first stage, these genes appear at that stage (see Section PIC-seq dataset in the main manuscript and S3 Table).
(PDF)

**S3 Table. Genes with high probabilities appearing in the topics identified in the reference.**
(PDF)

**S4 Table. Added as a separate spreadsheet: Top genes from the PIC-seq and the bone marrow datasets.**
(XLSX)

**S1 Fig. C145 cluster annotation: COVID-19 BALF, UMAP projection of patient sample C145.** We have identified the cluster containing doublets based on expression of marker genes and annotation by DoubletFinder.
(TIF)

**S2 Fig. Examples of genes for which we have not modified the expression.** As expected, the probability of observing their counts is similar under the two models.
(TIF)

**S3 Fig. Examples of genes for which we have modified the expression.** Their counts are observed with higher probability under the more complex model with new topics.
(TIF)

**S4 Fig. Additional synthetic experiment with a different set of randomly sampled genes modified.**
(TIF)

**S5 Fig. Additional synthetic experiment showing genes with unmodified expression.** Similar probabilities can be observed under the two models.
(TIF)

**S6 Fig. ROC curves for 50/50 doublets if the increase of counts for a set of genes is 3 or 10 respectively.**
(TIF)

**S7 Fig. 1$^{st}$ stage LDA with 5 topics, capturing topics specific to T-cells and DCs during the different timepoints.** No topic is shared across all cells. We fit a model with 5 topics on the initial reference population, co-culture of DCs and co-culture of T-cells. As can be seen from the figure, we are capturing topics that are unique to DCs and T-cells. For example, topic 2, seems to be expressed in T-cells at 20h, while topic 3 is expressed in DCs at 20h. However, we would expect at least some genes to be expressed across both T-cells and DCs, for example housekeeping or mitochondrial ones, and these do not appear to be represented by any topics. It seems possible therefore that one result of under-specifying is that some processes that

ought to be captured in stage 1 are actually captured in stage 2.
(TIF)

**S8 Fig. ROC curves for a range of topics.** Ground truth is considered the genes identified by [10]. Following 10 topics, which can be considered the optimum for this dataset, the performance starts to drop and the ROC curves for 40 and 100 topics show decay.
(TIF)

**S9 Fig. Area under the curve (AUC) is high for the optimal value of topics (10) and decays slowly after.**
(TIF)

**S10 Fig. Jensen-Shannon divergence and cosine follow a similar pattern to the perplexity.** JS increases from 10 (higher is better) and cosine decreases from 10 (lower is better).
(TIF)

**S11 Fig. New topics and PICs heatmap.** While some topics have higher probabilities associated with a timepoint, other topics appear expressed across all timepoints, so the gene expression shift is linked to the cell types that contribute to the PICs.
(TIF)

## Author Contributions

**Conceptualization:** Alexandrina Pancheva, Simon Rogers, Thomas D. Otto.

**Formal analysis:** Alexandrina Pancheva.

**Supervision:** Helen Wheadon, Simon Rogers, Thomas D. Otto.

**Visualization:** Alexandrina Pancheva.

**Writing – original draft:** Alexandrina Pancheva.

**Writing – review & editing:** Alexandrina Pancheva, Helen Wheadon, Simon Rogers, Thomas D. Otto.

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
