## [Decision Letter · Decision Letter 0]

27 Aug 2021

Dear MS. Pancheva,

Thank you very much for submitting your manuscript "Using topic modeling to detect cellular crosstalk in scRNA-seq" for consideration at PLOS Computational Biology.

As with all papers reviewed by the journal, your manuscript was reviewed by members of the editorial board and by several independent reviewers. In light of the reviews (below this email), we would like to invite the resubmission of a significantly-revised version that takes into account the reviewers' comments.

We cannot make any decision about publication until we have seen the revised manuscript and your response to the reviewers' comments. Your revised manuscript is also likely to be sent to reviewers for further evaluation.

Sincerely,

Qing Nie

Associate Editor

PLOS Computational Biology

Ilya Ioshikhes

Deputy Editor

PLOS Computational Biology

Reviewer's Responses to Questions

**Comments to the Authors:**

Reviewer #1: Comments to the Author

In this manuscript, the authors seek to use LDA as part of a procedure to identify genes that change their expression as a result of cell-cell interaction. Code is made available through the first author's Gihub. I found the application to be relevant and current, and the method to be interesting. However, I did think that the exposition of the approach needed tightening, and had a few other minor queries/comments, as described below.

Minor points:

1. The LDA model (Section 2.1). Typo: "\\alpha and \\beta are Dirichlet priors" -- I think these are vectors.

2. "Identifying topics linked to a cell type": for completeness and precision, it would be useful in this section to state the hypothesis that is being tested using Mann-Whitney's U.

3. In Table 1, why is 30 chosen as the lower bound for the number of topics? In general, how is an appropriate candidate set of values chosen?

4. Some of the language needs a little tightening, e.g. "Ranking genes potential candidates of interaction", "Let N be the doublets/interacting population of interest" (is N the sample size?), missing articles (e.g. "We expect the probability of observing counts greater or equal to the actual count for gene changed...", "Metadata file was used to filter ...", "Full list of filtering cut-offs..."), avoiding contractions (e.g. "a gene we haven’t modified...", "It's important to note...")

5. A 2-stage approach is adopted here, in which LDA is first performed on a reference population, and then another LDA is performed to try to identify new topics that capture any changes as result of interaction. I am not suggesting that the authors try this, but I wondered if they had instead considered a hierarchical model that would seek to model the two datasets jointly? If so, what would be the potential advantages/disadvantages of such an approach?

6. "Finding the number of topics" - here, average cosine distance is used, but are other options available? Why was this particular criterion chosen?

Reviewer #2: Pancheva et al. present a method for the analysis of cell-cell interactions through latent Dirichlet allocation (topic modeling). The methods introduced are interesting and I could see their potential impact when applied to current and emerging single-cell sequencing technologies. The paper in its current form is unclear in several places on the methods and the results presented, which limits comprehension of the main points being made. Specific concerns are listed below.

Major points

1. Definition of terms. I find the introduction of cell-cell communication via ligand-receptor interactions confusing, since ligands can in general diffuse, so might define a very different set of interacting cells than PICs. I urge the authors to be more careful / define more precisely what they mean by “an interaction”? In its current form, I don’t understand clearly when this refers to a general interaction between two cell types,. e.g. {T cell—Dendritic cell}, and when it refers to an interaction between/mediated by (the products of) two genes, e.g. cytokines/receptors/adhesion molecules, etc. This is an essential point as it is key to the paper. It also influences statements such as “do not allow for new interactions to be identified” (line 18) - cell/cell or gene/gene interactions?

2. Details on the two-step LDA procedure are not sufficient. Please explain the intuition behind this procedure in more detail (a diagram may help here). Specific questions: is it reasonable to assume that cells are not interacting in reference population? Surely some could have been/were interacting by chance before they were dissociated and sequenced? Use of the phrase “fixing topics” is not clear - as in: fixing the number of topics or fixing the entire distributions? “By fitting LDA on the co-cultures of one cell type in a dataset…” does this really mean one cell type? Or one sample of cells? My understanding was that you are fitting LDA for the _set_ of cell types that are present in the entire (non-interacting) cell population sample? Please explain.

3. For the number of topics: the example of table 1 is not clear. The cosine distances at 30 topics and 100 topics are very similar, implying potentially high sensitivity of # topics determined to the input data. Given the four distances calculated, it really begs the question: what are the cosine distances for topics < 30 and > 100?

4. “in the case of our synthetic experiments, we plot how choosing between 2 and 20 genes per topic affect the true positive and false positive rates” - where are these plots?

5. Why is adding 10 counts to the expression of a gene a good model for evidence that its expression is influenced by interaction — What is justification for this? Does the absolute change in counts used affect the results? More generally, more discussion of the results using synthetic data is needed. Figs 1 and 2 contain little information - what other plots would provide more detailed summaries of the inference results? e.g. (at least) plot more single genes than 2 in Fig 1; in fig 2 would be more helpful to provide total # mis-classified genes since the absolute numbers are small (ROC curves not v informative). Also: what is the effect of the abs count increase (if less than or greater than 10) on the results obtained? What is the detection limit? i.e. Can the method still pick up these interactions if they occur in 1 or 2 or 3 pairs of cells?

6. PIC-seq data: in Fig 3, it looks like there are some topics with very little variation across cells (e.g. topic 3), how to explain these? Also, I am curious what topics that pick up specific subsets of cells represent? e.g. topics 25 or 26. What are the top genes driving these? It would be helpful to include (supplementary) figures with genes contributing to topics for at least a good subset of the topics identified in Fig 3.

7. COVID-19 data: are the inconclusive results here due to: use of DoubletFinder + LDA not able to resolve interactions well (as the authors suggest), or due to the complexity/particularities of this dataset itself? Was DoubletFinder + LDA tried on any other dataset? In particular, there are a large number of clusters in the COVID dataset: perhaps trying DoubletFinder on a simpler (by # of cell types) 10X dataset would yield better results?

Minor points

- line 399: While this is informative, the current setup of the 10x Chromium protocol is fully suitable for studying cellular crosstalk of physically interacting cells” - is this missing “not”?

**Have the authors made all data and (if applicable) computational code underlying the findings in their manuscript fully available?**

Reviewer #1: Yes

Reviewer #2: Yes

PLOS authors have the option to publish the peer review history of their article (what does this mean?). If published, this will include your full peer review and any attached files.

Reviewer #1: No

Reviewer #2: No
---

## [Decision Letter · Decision Letter 1]

3 Dec 2021

Dear MS. Pancheva,

Thank you very much for submitting your manuscript "Using topic modeling to detect cellular crosstalk in scRNA-seq" for consideration at PLOS Computational Biology.

As with all papers reviewed by the journal, your manuscript was reviewed by members of the editorial board and by several independent reviewers. In light of the reviews (below this email), we would like to invite the resubmission of a significantly-revised version that takes into account the reviewers' comments.

The reviewer #2 still has major concerns. It's important in the re-submission that those concerns will be fully addressed. 

We cannot make any decision about publication until we have seen the revised manuscript and your response to the reviewers' comments. Your revised manuscript is also likely to be sent to reviewers for further evaluation.

Sincerely,

Qing Nie

Associate Editor

PLOS Computational Biology

Ilya Ioshikhes

Deputy Editor

PLOS Computational Biology

Reviewer's Responses to Questions

**Comments to the Authors:**

Reviewer #1: Thanks to the authors for largely addressing my points.

Two of the minor points I mentioned previously seem to have been missed:

1. The authors still assert that "α and β are Dirichlet priors that control..." -- but I think these are parameters of Dirichlet priors, not priors themselves.

2. Section title: "Ranking genes potential candidates of interaction" - I am not sure this currently makes sense. Is there a missing word?

Reviewer #2: In this revision the authors have addressed many of the concerns and the manuscript is improved for it. However, important questions remain, these are listed below.

1. Regarding the choice of number of topics, I understand that this is in general a challenge that perhaps no single metric can adequately quantify, but I still find the discussion in light of the metrics used (cosine distance and Jensen-Shannon divergence) to be inconclusive/not informative enough practically for potential users. For both these metrics the differences between a small and large number of topics is minimal. I am not sure what the observation of Fig. S5 adds? Is it not entirely expected that some of the topics from a many-topic model be contained (to some extent) in the topics of a few-topic model? Another question is: why did the average cosine differences in Table 1 change between the initial submission to the revision?

2. Practically, my concern is that there are no clear guidelines on choosing the number of topics for a user. Judging on cosine dist or JS, one would choose low (10) or low/high (10/150) numbers, respectively. Yet in examples, e.g. PIC-seq dataset, 30 topics are used. This may be due to the taking into account of “complexity” (which is not to my understanding well defined). For this method to be useful and adopted, this must be clarified. I think it will also be important to see some comparison of results with different numbers of topics, e.g. how do the heatmaps look for the PIC-seq dataset with 10 or 50 or 100 topics? These could be included as SI figs.

3. For new Fig 2, I appreciate the plots of additional genes in SI. However, the specific genes used are concerning, since the unmodified genes are almost all unrelated to immune cell phenotypes, i.e. characterizing general (housekeeping) cell processes (ribosome, actin). I think it is important to assess this in the analysis: switch the gene sets, and modify the expression of those in Fig S2, leaving the genes from Fig S3 unmodified, how would this affect the predictions of LDA?

4. Fig 3 please add total # genes predicted to the legend so the number of false positives can be understood in context.

5. For the PIC-seq data, thanks for the additional plots, they help with the analysis. A couple questions do arise. Why don’t the cells cluster by time point within type (T cell or DC) in Figs 4, S6? More importantly, I am still left wondering about the number of topics used here. In total, about 1/3 of the PIC-seq topics have been described by the authors, which is indeed helpful although some of them are not informative about immune processes as they contain mostly housekeeping genes. What are the remaining 2/3 of the topics characterizing? I would not hammer on this point so much were it not (as I understand it) the essential core of the methods. Thus I would like to understand if the other topics represent signal (if so what processes?) or background? If the latter, the method may still be useful but there needs to be some way to understand/characterize/rank(?) The topics.

**Have the authors made all data and (if applicable) computational code underlying the findings in their manuscript fully available?**

Reviewer #1: Yes

Reviewer #2: Yes

PLOS authors have the option to publish the peer review history of their article (what does this mean?). If published, this will include your full peer review and any attached files.

Reviewer #1: No

Reviewer #2: No
---

## [Decision Letter · Decision Letter 2]

25 Feb 2022

Dear MS. Pancheva,

We are pleased to inform you that your manuscript 'Using topic modeling to detect cellular crosstalk in scRNA-seq' has been provisionally accepted for publication in PLOS Computational Biology.

Best regards,

Qing Nie

Associate Editor

PLOS Computational Biology

Ilya Ioshikhes

Deputy Editor

PLOS Computational Biology

Reviewer's Responses to Questions

**Comments to the Authors:**

Reviewer #1: The authors have addressed my previous points.

Reviewer #2: I appreciate the authors diligence in addressing my questions and providing addition plots & support to justify the choice of the number of topics. In this revision all of my concerns have been addressed.

**Have the authors made all data and (if applicable) computational code underlying the findings in their manuscript fully available?**

Reviewer #1: Yes

Reviewer #2: Yes

PLOS authors have the option to publish the peer review history of their article (what does this mean?). If published, this will include your full peer review and any attached files.

Reviewer #1: No

Reviewer #2: No

---

## [Editor Report · Acceptance letter]

5 Apr 2022

PCOMPBIOL-D-21-01253R2 

Using topic modeling to detect cellular crosstalk in scRNA-seq

Dear Dr Pancheva,

I am pleased to inform you that your manuscript has been formally accepted for publication in PLOS Computational Biology. Your manuscript is now with our production department and you will be notified of the publication date in due course.

With kind regards,

Agnes Pap
